# AInstein: Assessing the Feasibility of AI-Generated Approaches to Research Problems

## Abstract

Large language models (LLMs) demonstrate impressive capabilities across a wide range of tasks, yet it remains unclear whether such success reflects genuine reasoning or sophisticated recall. We introduce **AInstein**, a framework for testing whether LLMs can generate valid solutions to AI research problems using *only* their pretrained parametric knowledge—without domain-specific fine-tuning, retrieval augmentation, or other external aids. Our approach extracts distilled problem statements from high-quality ICLR 2025 submissions, then tasks specialized solver agents with proposing and refining technical solutions through iterative critique loops, mimicking the cycles of proposal, review, and revision central to scientific inquiry. We evaluate AInstein on 1,214 ICLR papers stratified by acceptance tier (Oral, Spotlight, Poster), using an LLM-as-a-judge paradigm guided by a structured rubric, complemented by targeted manual checks. Performance is assessed with three metrics: *Success Rate* (does the solution address the problem?), *Rediscovery* (does it align with human-proposed methods?), and *Novelty* (does it yield valid, original approaches?). Our results reveal that while LLMs can rediscover feasible solutions and occasionally propose creative alternatives, their problem-solving ability remains fragile and highly sensitive to framing. These findings provide the first large-scale evidence on the extent to which LLMs can act as autonomous scientific problem-solvers, highlighting both their latent potential and their current limitations.

## 1 Introduction

The history of science is shaped by moments of profound insight: Newton's recognition that falling apples and orbiting planets obey the same laws, Darwin's theory of natural selection, Einstein's thought experiments on space and time. Each breakthrough required not only knowledge, but also the ability to **(a)** synthesize disparate observations into coherent theories and **(b)** traverse the conceptual path from problem to principle to solution. As artificial intelligence advances, we confront a critical question: do large language models (LLMs) exhibit such reasoning, or are they sophisticated pattern matchers relying on memorized associations (Xu et al., 2025; Chollet, 2019; Zhang et al., 2025; Min et al., 2022)?

This question is increasingly urgent as LLMs demonstrate strong performance in mathematics, programming, and even research workflows (Fang et al., 2024; Wang et al., 2024b; Zheng et al., 2025; Cui et al., 2025; Luo et al., 2025). Yet, much of this success may derive from associative recall rather than genuine conceptual reasoning. To probe this distinction, we ask the following question,

> **Q:** Can LLMs solve AI research problems using *only* their parametric knowledge?

Unlike prior evaluations that often test factual recall, benchmark performance (Luo et al., 2025; Chollet, 2019), or domain-specific competence, our study isolates the ability to generate solutions to open-ended research challenges without external aids such as fine-tuning or retrieval augmentation (Xu et al., 2025; Zheng et al., 2025; Cui et al., 2025)..

To that end, we introduce AInstein, a framework for empirically testing scientific problem-solving under these (Figure 1). The pipeline operates in two phases. In the *Problem Extraction Phase*, scientific abstracts are distilled into concise research challenges that preserve the core contribution while omitting direct references to the original solution. In the *Solution Phase*, solver agents generate and refine potential solutions through iterative critique loops, simulating the cycles of proposal, review, and revision characteristic of scientific inquiry. This design creates a controlled testbed for distinguishing recall from reasoning.

We curate a dataset of 1,214 high-quality ICLR 2025 papers, stratified by quality tier (Oral, Spotlight, Poster). Each paper serves as a source of a scientific abstract. For evaluation, we adopt an LLM-as-a-judge paradigm, where the LLM outputs are reviewed against a structured rubric that produce both structured scores and holistic judgments. To ensure reliability, we also manually check the solutions proposed by the LLM. We report the

Figure 1: The AINSTEIN framework. An input scientific abstract ($\mathcal{A}$) is first derived into a generalized problem ($\mathcal{P}$) by the Generalizer agent ($\mathcal{G}$). The Solver agent ($\mathcal{S}$) then attempts to derive a technical solution ($\mathcal{Z}$), using the problem statement $\mathcal{P}$. Both phases employ an iterative refinement loop with internal ($\mathcal{M}_i$) and external ($\mathcal{M}_e$) critique. **Note:** The transition $\mathcal{A} \rightarrow \mathcal{P}$ follows the same iterative refinement mechanism, where the the "Happy" condition is as described in Section 3.2.

three performance metrics for our task: **Success Rate**: does the proposed solution address the problem statement? **Rediscovery**: how often does an LLM independently converge on solutions that resemble those proposed by human researchers? **Novelty**: how often does the model generate approaches that are both valid and original (i.e., not proposed by humans)? Together, these metrics allow us to disentangle rote recall from genuine problem-solving and creativity.

Overall, this work makes three contributions. **First**, we propose a new evaluation paradigm that isolates scientific problem-solving ability by decoupling problem extraction from solution generation and removing external aids. **Second**, we introduce systematic metrics to measure success, rediscovery, and novelty, enabling a finer-grained analysis of reasoning versus recall. **Third**, we provide the first large-scale mapping of LLM problem-solving across models and research domains, revealing both promising signs of scientific creativity and the fragility of reasoning under different framings. Together, these contributions advance our understanding of LLMs not only as tools for research, but as potential autonomous problem-solvers.

## 2 RELATED WORK

The evaluation of scientific reasoning in language models has evolved along several complementary directions, each revealing different aspects of these systems' capabilities and limitations.

**Evaluating Scientific Knowledge and Reasoning.** Traditional benchmarks have focused on testing models' ability to answer scientific questions and solve domain-specific problems. Recent work has highlighted the limitations of these approaches, particularly their conflation of memorization with understanding. Wang et al. (2024a) demonstrate that models often rely heavily on pattern matching and retrieval rather than genuine reasoning when solving mathematical problems. Similarly, the phenomenon of "grokking"–where models suddenly transition from memorization to generalization–suggests that understanding and recall exist on a spectrum rather than as binary states (Power et al., 2022). Our work extends this line of inquiry by explicitly controlling for memorization through conceptual abstraction.

**Probing Model Representations.** Understanding how language models internally represent concepts has become a critical area of research. Investigations into models' latent knowledge have revealed that these systems often maintain multiple, sometimes contradictory, representations of the same concept (Longpre et al., 2021; Xie et al., 2023). The ability to "steer" model behavior through targeted interventions suggests that different representations can be selectively activated (Turner et al., 2023). Our framework leverages this insight, using varied problem framings to explore the full space of a model's conceptual representations.

**Emergence and Phase Transitions.** The study of emergent capabilities in large language models has revealed that certain abilities appear suddenly as models scale, resembling phase transitions in physical systems (Wei et al., 2022). Schaeffer et al. (2023) provide a nuanced analysis of these transitions, showing how measurement choices can create the appearance of emergence. Our concept of "conceptual activation energy" provides a complementary perspective, focusing on the minimum prompt complexity required to trigger specific reasoning capabilities rather than model scale.

**Scientific Discovery and Creativity.** Recent work has begun exploring whether AI systems can contribute to scientific discovery beyond mere assistance. Romera-Paredes et al. (2024) demonstrate that specialized systems can discover novel mathematical theorems, while other work has shown success in materials science and drug discovery (Zhavoronkov et al., 2019; Merchant et al., 2023). However, these systems typically operate within

narrow, well-defined domains with extensive domain-specific training. Our approach differs by testing general-purpose language models' ability to rediscover established concepts without domain-specific fine-tuning.

**Mechanistic Interpretability.** The growing field of mechanistic interpretability seeks to understand how models perform specific tasks by analyzing their internal computations (Elhage et al., 2021). This work has revealed that models can implement surprisingly sophisticated algorithms, from modular arithmetic to optimization procedures Olsson et al. (2022). Our framework complements these mechanistic studies by providing behavioral evidence of models' algorithmic capabilities through their ability to rediscover solutions.

**Prompt Engineering and In-Context Learning.** The sensitivity of language models to prompt formulation has been extensively documented, with small changes in wording capable of drastically altering performance (Reynolds & McDonell, 2021). Work on in-context learning has shown that models can adapt to new tasks through careful example selection and prompt design (Brown et al., 2020; Min et al., 2022). We build on these insights, treating prompt design not as an engineering challenge but as a scientific tool for probing models' latent capabilities.

Our work synthesizes insights from these diverse research threads while introducing a novel evaluation paradigm. Rather than testing what models know or can retrieve, we examine whether they can recreate the process of discovery itself. This approach provides a new lens for understanding the relationship between memorization, pattern matching, and genuine reasoning in artificial intelligence systems.

## 3 METHODOLOGY

We present AINSTEIN, a framework for evaluating whether large language models (LLMs) can act as autonomous scientific problem-solvers. The framework is designed to (1) extract concise research problems from scientific abstracts, and (2) test whether solver agents can propose technically valid solutions through iterative refinement. Our approach formalizes this process as a structured pipeline with nested critique loops, which has been shown to improve LLMs' problem-solving capability (Shinn et al., 2023; Madaan et al., 2023).

### 3.1 PROBLEM-TO-SOLUTION PIPELINE

The AINSTEIN pipeline consists of two distinct phases, separating problem formulation from solution generation to create a controlled testbed for probing scientific reasoning.

**Phase 1: Problem Extraction.** Given a scientific abstract $\mathcal{A}$, a *Generalizer* agent $\mathcal{G}$ (an LLM prompted with a specific role) produces a distilled problem statement $\mathcal{P}$. The central goal is to create a high-quality problem formulation that is both faithful to the original challenge and free of solution-specific artifacts. To this end, $\mathcal{G}$ is explicitly instructed to maximize **abstraction** and **fidelity** while minimizing **ambiguity** and **solution leakage** (as defined in our evaluation criteria). This constraint is critical to prevent the agent from simply paraphrasing the solution from the abstract, thereby ensuring the subsequent solving task is non-trivial. Formally, the process is: $\mathcal{P} = \mathcal{G}(\mathcal{A}; \mathcal{M}_i, \mathcal{M}_e)$.

**Phase 2: Solution Generation.** A *Solver* agent $\mathcal{S}$ (a separate LLM instance) receives only the problem statement $\mathcal{P}$ and proposes a technical solution $\mathcal{Z}$. This phase evaluates the model's ability to generate a detailed and potentially novel methodology to address the problem, without access to the original abstract or solution. Formally: $\mathcal{Z} = \mathcal{S}(\mathcal{P}; \mathcal{M}_i, \mathcal{M}_e)$.

### 3.2 ITERATIVE REFINEMENT VIA CRITIQUE LOOPS

Both the Generalizer and Solver operate under a nested refinement mechanism that mimics scientific inquiry: iterative proposal, critique, and revision, shown in Algorithm 1. Each phase has:

1. **Internal Critique Loop (Model $\mathcal{M}_i$):** The inner loop simulates fast, low-cost self-correction. An initial draft (problem statement or solution) is generated, then reviewed against task-specific criteria by an internal critic (also $\mathcal{M}_i$). The critic provides rubric-based feedback and a binary accept/reject signal. If rejected, the draft is revised. This process repeats for up to `MaxInternalAttempts` iterations.

2. **External Critique Loop (Model $\mathcal{M}_e$):** The outer loop provides a higher-fidelity review from a stronger external model $\mathcal{M}_e$. Once the internal loop converges, $\mathcal{M}_e$ evaluates the candidate artifact. If the external judgment is negative, the feedback from both critics is incorporated into a new attempt, up to `MaxExternalAttempts`. This ensures that only high-quality outputs are accepted.

This dual-loop structure operationalizes the dynamics of scientific peer review: rapid local iteration combined with stricter external scrutiny.

---

**Algorithm 1** The AInstein Algorithm

---

1: **Input:** Abstract $\mathcal{A}$, Internal model $\mathcal{M}_i$, External model $\mathcal{M}_e$
2: **Output:** Problem $\mathcal{P}_{final}$, Solution $\mathcal{Z}_{final}$
3: $\mathcal{P}_{final}, \mathcal{P}_{candidate} \leftarrow$ null
4: **for** $e = 1$ **to** MaxExternalAttempts **do**
5:  **for** $i = 1$ **to** MaxInternalAttempts **do**
6:   Generate $\mathcal{P}_{candidate}$; self-critique($\mathcal{P}_{candidate}$)
7:   **if** pass **then break**
8:   **end if**
9:  **end for**
10:  External critique($\mathcal{P}_{candidate}$)
11:  **if** pass **then** $\mathcal{P}_{final} \leftarrow \mathcal{P}_{candidate}$; **break**
12:  **end if**
13: **end for**
14: **if** $\mathcal{P}_{final} \neq$ null **then**
15:  $\mathcal{Z}_{final} \leftarrow$ SolveWithRefinement($\mathcal{P}_{final}, \mathcal{M}_i, \mathcal{M}_e$)
16: **end if**

---

## 4 EXPERIMENTAL SETUP

### 4.1 DATASETS & MODELS

Our experimental dataset consists of 1,214 papers curated from the ICLR 2025 conference submissions, drawing inspiration from the ICLR Dataset (González-Márquez & Kobak, 2024). This corpus provides a large, diverse, and contemporary set of high-quality research problems. Our curated set is intentionally stratified by the papers' final acceptance tiers: **Oral**, **Spotlight**, and **Poster**. Table 1 compares the distribution of our final dataset against the full ICLR 2025 corpus, highlighting how our curation process oversamples higher-quality papers to analyze how model performance correlates with the perceived quality and impact of the research.

To prevent data leakage and ensure our evaluation tests reasoning rather than retrieval, all models used in our experiments have knowledge cutoffs that predate the ICLR 2025 submission deadline. We experiment with three primary model families selected to represent different scales of capability. The large-scale models include `GPT-OSS-120B` and `Qwen-235B`, representing the state-of-the-art in reasoning. The mid-scale model is `Mistral-24B`, a powerful and widely used alternative.

Table 1: Comparison of paper distribution between the full ICLR 2025 dataset and our curated experimental set (N=1,214).

| Decision Tier | Full ICLR Dataset | | Our Curated Set | |
|---|---|---|---|---|
| | Count | % | Count | % |
| Oral | 213 | 1.8 | 213 | 17.5 |
| Spotlight | 379 | 3.2 | 379 | 31.2 |
| Poster | 3,111 | 26.7 | 622 | 51.2 |
| Rejected | 5,014 | 43.0 | — | — |
| Withdrawn | 2,946 | 25.3 | — | — |
| **Total** | **11,663** | **100** | **1,214** | **100** |

It is important to distinguish between the *task roles* (Generalizer and Solver agents) and the *refinement roles* (Internal Model $\mathcal{M}_i$ and External Model $\mathcal{M}_e$). We systematically test all pairings to analyze the interplay between model capabilities for generation and critique, avoiding biases that might arise from using models of the same family or scale in both roles.

### 4.2 EVALUATION

We use GPT-OSS-120B as an LLM-as-a-judge, which is known to correlate well with human evaluation (Liu et al., 2023; Sottana et al., 2023). For each generated problem statement (P), we calculate a deficit score (d) based on four criteria: *Fidelity* to the original challenge, *Abstraction* from implementation details, lack of *Ambiguity*, and avoidance of solution *Leakage*. A lower deficit score indicates a higher-quality problem formulation. For each solution (Z), the judge provides a 1-5 score based on its technical *Feasibility*, *Completeness*, and *Novelty*. This score informs our three primary metrics: **(1) Success Rate:** Is the solution both feasible and complete? **(2) Rediscovery:** Does a successful solution match the original human solution? **(3) Novel & Valid:** Is a successful solution valid but different from the original?

**Quantitative Validation Metrics.** The core metrics are complemented by quantitative text analyses. For semantic coherence, we generate 4096-dimensional embeddings using `Qwen3-Embedding-8B`. To optimize for the target relationship (e.g., problem-solution alignment vs. end-to-end rediscovery), we prepend each text pair with a task-specific instruction before embedding. From these embeddings, we compute Cosine Similarity as our primary metric, along with Euclidean distance. We also assess textual complexity using standard readability scores (e.g., Flesch-Kincaid Grade Level).

**Competitive Ranking via Human-Verified ELO.** To provide a robust aggregate ranking, we conduct a head-to-head tournament where human evaluators provide pairwise preferences between solutions. From these preferences, we compute an ELO rating for each agent configuration.

# 5 RESULTS

Our experimental results reveal a clear hierarchy of scientific reasoning capabilities among model configurations and offer nuanced insights into the nature of machine-driven discovery. We structure our analysis around three key findings: (1) the dominant role of the internal model ($\mathcal{M}_i$) in determining success, (2) the distinction between precise rediscovery and creative problem-solving, and (3) the validation of these findings through competitive ranking, quantitative metrics and qualitative study.

**Problem Generalization Analysis** The validity of our study hinges on the quality of the abstracted problems given to the Solver agents. To analyze this, we first evaluate the performance of our three models in the Generalizer role. The correlation matrix in Figure 2 empirically validates our formulation, confirming that our deficit score is strongly correlated with these key quality dimensions.

As shown in Table 2, both `Qwen-235B` and `GPT-OSS-120B` produce problem statements with very low deficit scores ($d \approx 2.5$), while `Mistral-24B`'s are of slightly lower quality ($d \approx 3.5$). Statistical significance testing between GPT and OSS confirms that the two models perform comparably (see Appendix B). Thus, we use the outputs from all three Generalizers as inputs to the solution phase for a complete study. This ensures our Solver results are robust and not overfitted to a single problem-framing style, effectively testing each Solver's adaptability.

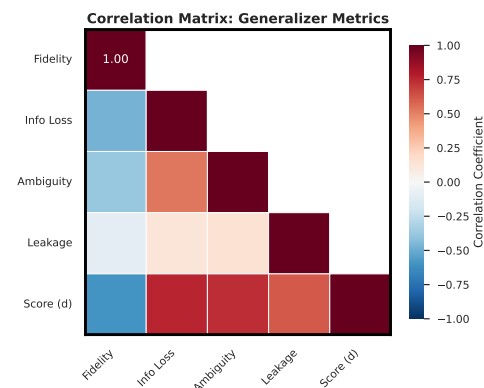

Figure 2: Correlation matrix for Generalizer quality metrics. The deficit score ($d$) is strongly correlated with Information Loss and Ambiguity.

Table 2: Generalizer metrics by model configuration. The deficit score ($d \downarrow$) quantifies quality degradation as the average penalty across four dimensions (Fidelity, Info Loss, Ambiguity, and Leakage). The large-scale models demonstrate superior performance.

| Model Configuration | | Fidelity ↑ | Info Loss ↓ | Ambiguity ↓ | Leakage ↓ | $d \downarrow$ |
|---|---|---|---|---|---|---|
| **External Model** | **Internal Model** | | | | | |
| GPT-OSS-120B | GPT-OSS-120B | 8.80 ± 0.77 | 3.72 ± 1.24 | 3.08 ± 1.11 | 2.16 ± 1.55 | 2.54 ± 0.76 |
| | Mistral-24B | 8.39 ± 0.85 | 5.24 ± 1.28 | 5.41 ± 0.85 | 1.96 ± 1.45 | 3.56 ± 0.70 |
| | Qwen-235B | 8.86 ± 0.49 | 4.24 ± 1.92 | 3.03 ± 1.40 | 1.68 ± 1.31 | 2.52 ± 0.94 |
| Mistral-24B | GPT-OSS-120B | 8.82 ± 0.66 | 3.70 ± 1.16 | 3.05 ± 1.03 | 2.19 ± 1.57 | 2.53 ± 0.70 |
| | Mistral-24B | 8.41 ± 0.82 | 5.16 ± 1.30 | 5.39 ± 0.80 | 1.89 ± 1.39 | 3.51 ± 0.66 |
| | Qwen-235B | 8.88 ± 0.47 | 4.19 ± 1.91 | 3.07 ± 1.42 | 1.65 ± 1.23 | 2.51 ± 0.93 |
| Qwen-235B | GPT-OSS-120B | 8.79 ± 0.80 | 3.71 ± 1.24 | 3.06 ± 1.12 | 2.17 ± 1.56 | 2.54 ± 0.76 |
| | Mistral-24B | 8.41 ± 0.85 | 5.18 ± 1.34 | 5.39 ± 0.87 | 1.97 ± 1.54 | 3.53 ± 0.74 |
| | Qwen-235B | 8.87 ± 0.46 | 4.23 ± 1.91 | 3.02 ± 1.32 | 1.63 ± 1.22 | 2.50 ± 0.89 |

**The Internal Model's Capability is Paramount.** Across all experiments, the single most predictive factor of success is the strength of the internal model, $\mathcal{M}_i$. The results in Table 3 show a clear performance stratification: configurations using `GPT-OSS-120B` as the internal agent consistently and substantially outperform all others. For instance, when solving problems from the `GPT-OSS-120B` generalizer, the `GPT-OSS-120B` internal agent achieves a Success Rate of 74.05% under strict evaluation, whereas the next-best model, `Qwen-235B`, reaches only 43.82%.

**Rediscovery vs. Creative Problem-Solving.** Our framework reveals a critical distinction between reproducing an existing solution and discovering a novel, valid one. A comparison between relaxed ($\tau = 4$) and strict ($\tau = 5$) evaluation thresholds, presented in Table 3, is particularly telling. Under the relaxed threshold, the top `GPT-OSS-120B` agent achieves high Rediscovery rates (75-84%), suggesting it often generates solutions

conceptually close to the original - successful "near misses." However, when tightening the criterion to demand functional equivalence ($\tau = 5$), these scores plummet to 15-20%. The stark drop reveals that perfect rediscovery is exceptionally rare. In contrast, the *Novel & Valid* metric for the same agent remains high and remarkably stable across both thresholds. This indicates that when models do not perfectly rediscover the solution, they often find *alternative*, sound scientific approaches.

Table 3: LLM-as-a-judge success rates (%) comparing lenient ($\tau = 4$) and strict ($\tau = 5$) evaluation thresholds. Bold indicates the best-performing internal model for each problem source and external critic combination.

| Model Configuration | | Rediscovery | | SR Solver | | Novel & Valid | |
|---|---|---|---|---|---|---|---|
| External ($\mathcal{M}_e$) | Internal ($\mathcal{M}_i$) | $\tau = 4$ | $\tau = 5$ | $\tau = 4$ | $\tau = 5$ | $\tau = 4$ | $\tau = 5$ |
| **Problem Source: GPT-OSS-120B** | | | | | | | |
| GPT-OSS-120B | **GPT-OSS-120B** | **83.77** | **19.11** | **92.34** | **74.05** | **74.46** | **59.39** |
| | Mistral-24B | 65.73 | 6.43 | 75.45 | 34.60 | 70.10 | 31.80 |
| | Qwen-235B | 69.03 | 7.74 | 80.48 | 43.82 | 73.56 | 40.20 |
| Mistral-24B | **GPT-OSS-120B** | **83.94** | **17.79** | **93.41** | **71.91** | **76.44** | **58.98** |
| | Mistral-24B | 68.37 | 6.75 | 74.46 | 31.22 | 68.86 | 29.00 |
| | Qwen-235B | 72.49 | 7.17 | 85.34 | 39.21 | 79.08 | 36.74 |
| Qwen-235B | **GPT-OSS-120B** | **80.89** | **17.38** | **88.63** | **70.35** | **71.99** | **57.74** |
| | Mistral-24B | 65.98 | 7.08 | 76.52 | 34.84 | 70.43 | 31.63 |
| | Qwen-235B | 70.51 | 7.66 | 85.34 | 42.34 | 79.00 | 39.87 |
| **Problem Source: Mistral-24B** | | | | | | | |
| GPT-OSS-120B | **GPT-OSS-120B** | **76.44** | **16.56** | **92.83** | **78.01** | **76.94** | **64.58** |
| | Mistral-24B | 54.12 | 7.58 | 80.97 | 42.42 | 74.30 | 38.30 |
| | Qwen-235B | 61.53 | 8.40 | 84.68 | 50.16 | 76.94 | 45.14 |
| Mistral-24B | **GPT-OSS-120B** | **76.19** | **16.97** | **93.33** | **79.41** | **77.10** | **66.14** |
| | Mistral-24B | 55.52 | 8.15 | 81.80 | 40.86 | 74.88 | 37.07 |
| | Qwen-235B | 64.09 | 7.91 | 87.73 | 49.09 | 80.56 | 44.89 |
| Qwen-235B | **GPT-OSS-120B** | **74.79** | **15.57** | **89.21** | **76.28** | **74.55** | **63.51** |
| | Mistral-24B | 59.14 | 8.73 | 83.11 | 43.57 | 75.54 | 39.04 |
| | Qwen-235B | 64.00 | 9.14 | 87.48 | 49.84 | 79.24 | 44.73 |
| **Problem Source: Qwen-235B** | | | | | | | |
| GPT-OSS-120B | **GPT-OSS-120B** | **77.68** | **14.91** | **90.86** | **77.51** | **76.52** | **65.24** |
| | Mistral-24B | 58.73 | 5.11 | 78.01 | 41.85 | 73.81 | 39.21 |
| | Qwen-235B | 64.58 | 6.43 | 83.53 | 50.25 | 77.92 | 46.79 |
| Mistral-24B | **GPT-OSS-120B** | **76.77** | **14.74** | **91.68** | **77.10** | **78.17** | **66.56** |
| | Mistral-24B | 58.57 | 4.37 | 79.57 | 38.88 | 76.19 | 36.49 |
| | Qwen-235B | 66.64 | 6.84 | 87.56 | 49.75 | 81.30 | 46.13 |
| Qwen-235B | **GPT-OSS-120B** | **76.85** | **14.09** | **91.43** | **77.18** | **78.17** | **66.89** |
| | Mistral-24B | 60.05 | 6.34 | 80.89 | 41.02 | 75.86 | 37.97 |
| | Qwen-235B | 65.82 | 6.75 | 87.64 | 50.91 | 81.38 | 47.20 |

**Performance Across Paper Tiers** Our stratification of the dataset by paper quality (Oral, Spotlight, Poster) allows us to investigate whether LLM problem-solving ability correlates with the perceived difficulty of the research. As illustrated in Figure 3, the performance hierarchy across internal models remains remarkably consistent across all three tiers. The top-performing GPT-OSS-120B agent shows stable Success Rates for Oral (69.0%), Spotlight (77.8%), and Poster (72.5%) papers. This counter-intuitive result suggests that the model's ability to generate a valid solution is not significantly hindered by the novelty or impact of the original paper.

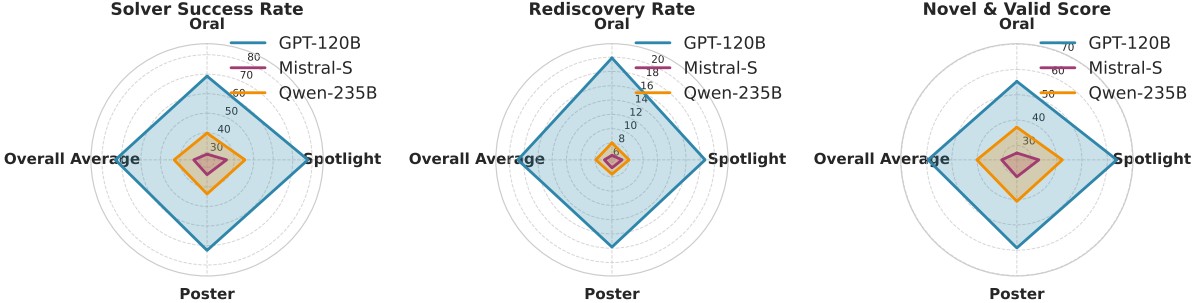

Figure 3: Performance comparison of internal models across ICLR paper tiers (Oral, Spotlight, Poster), averaging across GPT-OSS-120B as the external model. The results are shown for the strict threshold ($\tau = 5$). The performance hierarchy is stable, and the top model (GPT-OSS-120B) is not significantly impacted by the paper's prestige.

**Robustness to Evaluator Choice.** A critical concern in our study is the potential for the LLM judge to introduce bias. To verify that our findings are not an artifact of a specific evaluator, we replaced our judge (GPT-OSS-120B) with Qwen3-235B and replicated the LLM-as-a-judge scores. The results, presented in Table 4, confirm that our central conclusions are robust. We observe the same sharp decline in Rediscovery rates when moving from a lenient ($\tau = 4$) to a strict ($\tau = 5$) evaluation threshold, regardless of the judge as well as the performance from GPT-OSS-120B continues to outperform the others.

Table 4: LLM-as-a-judge success rates (%) using **Qwen3-235B as the evaluator**, comparing lenient ($\tau = 4$) and strict ($\tau = 5$) thresholds. This table validates that the findings from the primary evaluator (Table 3) are robust. Bold indicates the best-performing internal model for each external critic.

| Model Configuration | | Rediscovery | | SR Solver | | Novel & Valid | |
|---|---|---|---|---|---|---|---|
| External ($\mathcal{M}_e$) | Internal ($\mathcal{M}_i$) | $\tau = 4$ | $\tau = 5$ | $\tau = 4$ | $\tau = 5$ | $\tau = 4$ | $\tau = 5$ |
| GPT-OSS-120B | **GPT-OSS-120B** | **77.17** | **15.68** | **97.67** | **94.25** | **79.81** | N/A |
| | Mistral-24B | 62.58 | 6.54 | 92.95 | 70.81 | 66.86 | N/A |
| | Qwen-235B | 64.92 | 7.30 | 98.49 | 86.82 | 80.77 | N/A |
| Mistral-24B | **GPT-OSS-120B** | **76.86** | **14.61** | **98.31** | **93.07** | **80.07** | N/A |
| | Mistral-24B | 61.06 | 5.63 | 93.54 | 70.59 | 66.61 | N/A |
| | Qwen-235B | 62.71 | 6.77 | 98.08 | 85.62 | 79.93 | N/A |
| Qwen-235B | **GPT-OSS-120B** | **76.84** | **14.62** | 98.97 | **94.53** | 80.94 | N/A |
| | Mistral-24B | 63.61 | 7.44 | 93.88 | 75.10 | 69.73 | N/A |
| | Qwen-235B | 64.24 | 7.10 | 98.48 | 88.25 | 82.25 | N/A |

**Human-Verified Competitive Ranking.** To move beyond automated metrics and validate our findings with human judgment, a head-to-head tournament was conducted where the authors of this study served as human evaluators. To mitigate potential bias, solutions from competing agent configurations were presented in a randomized and anonymized fashion. From these pairwise preferences, we computed an ELO rating for each configuration (Table 5). This human-verified ranking establishes a definitive hierarchy, with the GPT-OSS-120B self-play configuration emerging as the clear top performer (ELO 1119), second only to the ground-truth human abstracts. This result robustly confirms the primacy of the internal model's strength. We present detailed case studies in our Appendix C.

Table 5: ELO ratings derived from a head-to-head tournament where human evaluators provided pairwise preferences between solutions.

| Agent | ELO Rating | Wins | Losses | Win Rate (%) |
|---|---|---|---|---|
| Human Abstracts | 1187 | 17 | 1 | 94.4 |
| **GPT-OSS-120B + GPT-OSS-120B** | **1119** | **14** | **5** | **63.6** |
| Mistral-24B + GPT-OSS-120B | 939 | 5 | 8 | 25.0 |
| Mistral-24B + Mistral-24B | 927 | 5 | 12 | 25.0 |
| GPT-OSS-120B + Mistral-24B | 828 | 1 | 16 | 5.0 |

---

**Case Study: Human-Verified ELO Evaluation**

**Problem:** How can an online RL agent learn in non-stationary environments and prevent catastrophic forgetting, given only a continuous stream of experience without task labels?

**Agent A** (Mistral-24B Self-Play): We propose a **Context-Aware Meta-Reinforcement Learning with Dual Memory (CAM-RL-DM)** architecture, which integrates a meta-learning component with dual episodic and semantic memory modules. The meta-learner rapidly adapts to new contexts by learning a context-encoding function that parameterizes the policy network...

**Agent B** (GPT-OSS-120B Self-Play): We introduce **Contextual Continual Actor-Critic (CCAC)**, an online RL architecture that (i) encodes the recent trajectory into a low-dimensional context vector, (ii) routes state-action decisions through a mixture-of-experts policy where gating weights are computed by a softmax over the context, and (iii) augments each expert with a lightweight generative replay model...

**Verdict: Agent B wins.** The evaluator noted that Agent B's proposal of a Mixture-of-Experts policy with generative replay was a more concrete and powerful mechanism for preventing catastrophic forgetting than Agent A's more general dual-memory system.

---

**Quantitative Text Analysis.** We further validate our findings with quantitative text analysis (Table 6). Semantic coherence metrics show that solutions generated by the top-performing internal model, GPT-OSS-120B, consistently achieve high cosine similarity scores ($\approx 0.87$) with the problems they address. This indicates a strong conceptual alignment between the generated solutions and the research challenges. Furthermore, readability scores reveal a clear distinction in textual complexity. The large-scale models, GPT-OSS-120B and Qwen-235B, produce solutions with a significantly higher Flesch-Kincaid grade level ($\approx 23 - 26$) compared to the mid-scale Mistral-24B ($\approx 22$). This suggests that the more capable models generate more technically dense and linguistically sophisticated proposals, which aligns with their superior performance on our core reasoning metrics.

Table 6: Semantic distance and readability metrics by model configuration. Metrics are mean ± std.

| Model Configuration | | Semantic Metrics | | Readability Metrics | |
| --- | --- | --- | --- | --- | --- |
| External Model | Internal Model | Cosine Sim. ↑ | Euclidean ↓ | Flesch Ease ↑ | Flesch Grade ↓ |
| | GPT-OSS-120B | 0.868 ± 0.035 | 0.509 ± 0.067 | -11.04 ± 18.28 | 22.74 ± 4.4 |
| GPT-OSS-120B | Mistral-24B | 0.852 ± 0.054 | 0.536 ± 0.098 | -2.81 ± 17.63 | 21.52 ± 4.03 |
| | Qwen-235B | 0.863 ± 0.041 | 0.518 ± 0.077 | -24.21 ± 18.93 | 25.56 ± 4.17 |
| | GPT-OSS-120B | 0.873 ± 0.035 | 0.499 ± 0.069 | -10.25 ± 17.91 | 22.72 ± 4.5 |
| Mistral-24B | Mistral-24B | 0.867 ± 0.05 | 0.507 ± 0.094 | -4.26 ± 17.65 | 22.07 ± 4.22 |
| | Qwen-235B | 0.869 ± 0.04 | 0.507 ± 0.077 | -25.42 ± 20.31 | 25.86 ± 4.98 |
| | GPT-OSS-120B | 0.874 ± 0.035 | 0.498 ± 0.068 | -10.76 ± 18.93 | 22.81 ± 4.7 |
| Qwen-235B | Mistral-24B | 0.862 ± 0.052 | 0.515 ± 0.097 | -3.26 ± 17.9 | 21.8 ± 4.15 |
| | Qwen-235B | 0.868 ± 0.041 | 0.508 ± 0.077 | -24.7 ± 20.32 | 25.7 ± 4.84 |

**Qualitative Text Analysis.** To understand the conceptual landscape of the generated solutions, we performed a qualitative analysis by categorizing the entire solution space. We first grouped all solutions into semantically coherent clusters via KMeans++ on their `Qwen-8B` embeddings. Each cluster was then assigned an interpretable label using its most characteristic TF-IDF keywords.

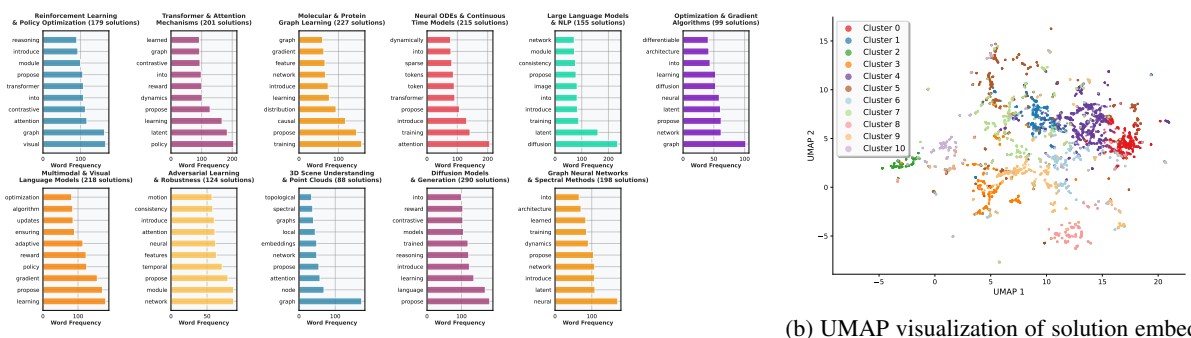

(a) Keyword frequency for eight prominent clusters.

(b) UMAP visualization of solution embeddings.

Figure 4: Visual analysis of the 11 identified research clusters. (a) Top keywords provide a thematic summary for several clusters. (b) The UMAP projection illustrates the distinct grouping of solutions, with each color corresponding to a unique thematic cluster.

This process reveals a rich taxonomy of 11 distinct solution archetypes, summarized in Table 8 and visualized in Figure 4. The analysis demonstrates that the agents do not merely produce generic outputs but explore a diverse range of modern research paradigms. These archetypes span broad strategies like Reinforcement Learning (C0), foundational model architectures and Transformers (C1), and highly specialized domains such as Molecular Graph Modeling (C2) and 3D Scene Representation (C8). We detail representative solutions for these clusters in our Appendix D.

## 6 CONCLUSION

In this work, we investigated whether large language models can function as autonomous scientific problem-solvers, using only their parametric knowledge. To this end, we introduced AINSTEIN, a novel framework that extracts core research challenges from scientific abstracts and tasks LLM agents with generating and refining technically sound solutions through iterative critique. Our evaluation yielded reveals two key findings: a solution's quality is overwhelmingly determined by the core reasoning model's capability, and models excel at generating novel, scientifically valid alternatives rather than perfectly rediscovering human solutions. These results provide compelling evidence that modern LLMs possess latent scientific reasoning abilities that go beyond sophisticated pattern matching. Our work is primarily focused on the AI domain, and the LLM-as-a-judge paradigm carries inherent biases; future work should therefore extend this framework to other scientific fields like biology and physics, explore more sophisticated methods for problem extraction from full-text articles, and investigate how different agent architectures could further unlock these nascent problem-solving capabilities.

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

APPENDIX

## A  LLM USAGE

We have used LLMs to polish the text of the paper and also to conduct literature review. Specifically, we used chatGPT's deep research feature to retrieve some relevant papers.

## B  STATISTICAL SIGNIFICANCE TESTING

To rigorously compare the performance between our primary models, GPT-OSS-120B and Mistral-24B, we performed two-sample t-tests and non-parametric Mann-Whitney U tests. We applied a Bonferroni correction to account for multiple comparisons, resulting in a corrected significance threshold of $\alpha \approx 0.0036$. The results are summarized in Table 7. While we find statistically significant differences in readability metrics, for the core Generalizer metrics, we find no significant differences. We conclude that while GPT-OSS-120B may have a slight, statistically detectable edge, the two models perform at a broadly comparable level in our framework. Thus, motivating our choice of considering problem statements from multiple sources.

## C  ADDITIONAL HUMAN EVALUATION EXAMPLES

This appendix provides a selection of head-to-head comparisons from our human-verified ELO tournament. For each example, we present the problem statement, the responses from the two competing agents, and the final verdict from the human evaluator.

Table 7: Significance testing for key metrics comparing Mistral-24B (M) and GPT-OSS-120B (G). An asterisk (*) denotes statistical significance at the Bonferroni-corrected level ($\alpha \approx 0.0036$). Despite some statistical significance, all effect sizes are negligible.

| Category | Metric | Mean (M) | Mean (G) | $p$ (t-test) | $p$ (MW-U) | Cohen's d |
|---|---|---|---|---|---|---|
| Generalizer | Semantic Fidelity | 8.610 | 8.600 | 0.468 | 0.898 | 0.021 |
| | Information Loss | 4.430 | 4.480 | 0.254 | 0.288 | $-0.033$ |
| | Ambiguity | 4.220 | 4.250 | 0.601 | 0.659 | $-0.015$ |
| | Solution Leakage | 2.040 | 2.060 | 0.602 | 0.207 | $-0.015$ |
| Readability | Flesch-Kincaid Grade* | 22.270 | 21.970 | 0.002 | 0.001 | 0.070 |

**Example 1: Video Inverse Problems (ID: TRWxFUzK9K)**

- **Problem Statement:** How can we formulate and solve video inverse problems that involve spatio-temporal degradations using only pre-trained image diffusion models, while ensuring consistent reconstruction across the temporal dimension without training dedicated video diffusion models?

- **Agent A Response (`Mistral-24B` Self-Play):** We propose a method called **Temporal Consistency-aware Recurrent Image Diffusion (TC-RID)** that adapts pre-trained image diffusion models for video inverse problems. TC-RID introduces a recurrent neural network (RNN) that processes each video frame sequentially, using the hidden state to maintain temporal information...

- **Agent B Response (`GPT-OSS-120B` Self-Play):** We formulate video inverse problems as a sequence of frame-wise diffusion processes in which a frozen pre-trained image diffusion model acts as the denoiser, but each denoising step receives an additional cross-frame guidance tensor obtained by warping the latent of the previous frame with a fast optical-flow estimator...

- **Winner: B**. Agent B's solution was judged to be more technically specific and innovative by proposing the use of optical flow and warped latents as a temporal prior.

**Example 2: Dynamic Mixture of Experts (ID: T26f9z2rEe)**

- **Problem Statement:** How can a Mixture of Experts architecture automatically determine the appropriate number of experts to activate for each input token and dynamically adjust the total number of experts during training, thereby eliminating the need for extensive hyper-parameter search?

- **Agent A Response (Human Abstract):** The Sparse Mixture of Experts (SMoE) has been widely employed... However, the performance of SMoE heavily depends on the choice of hyper-parameters... As a remedy, we introduce the Dynamic Mixture of Experts (DynMoE) technique. DynMoE incorporates (1) a novel gating method that enables each token to automatically determine the number of experts to activate...

- **Agent B Response (`GPT-OSS-120B` Self-Play):** We introduce **Dynamic Adaptive Gating with an Elastic Expert Pool (DAGE-E)**: a global gating network outputs a full softmax over a large superset of experts, while a learned token-wise budget scalar determines a differentiable top-k cutoff... Concurrently, an Elastic Expert Pool manager monitors per-expert utilization statistics...

- **Winner: A (Human)**. While the LLM's proposal (DAGE-E) was highly detailed and technically plausible, the human abstract was judged to be a slightly more direct and elegant solution to the core problem.

**Example 3: Configurable LLM Safety (ID: ERce2rgMQC)**

- **Problem Statement:** How can large language models be equipped with a mechanism that allows their safety behavior to be dynamically configured at inference time to satisfy diverse cultural, regional, and individual safety requirements, without requiring model retraining?

- **Agent A Response (`GPT-OSS-120B` Self-Play):** We introduce a **Dynamic Safety Router (DSR)** that sits between the transformer layers of a frozen LLM and its output head... the DSR computes per-token gating vectors that multiplicatively modulate the hidden state...

- **Agent B Response (`Mistral-24B` → `GPT-OSS-120B`):** We introduce a **Safety-Conditioned HyperAdapter (SCH)** architecture that sits between a frozen LLM and its output: a lightweight hypernetwork receives a structured safety profile and instantly emits low-rank additive weight matrices for each transformer layer...

- **Winner: B**. The HyperAdapter approach was judged to be a more modern and flexible architecture for inference-time configuration compared to the gating mechanism.

### Example 4: Accelerating Diffusion Models (ID: OlzB6LnXcS)

- **Problem Statement:** How can we substantially accelerate the sampling process of diffusion and flow-matching generative models while preserving or improving sample quality, using a single network trained in a single phase?
- **Agent A Response (`Mistral-24B` Self-Play):** We propose a novel architecture, **Sparse Adaptive Flow Network (SAFNet)**, which integrates adaptive sparse convolutions with dynamic skip connections...
- **Agent B Response (`GPT-OSS-120B` Self-Play):** We introduce **Dynamic-Step Diffusion (DSD)**: a continuous-time diffusion model that receives... a learned embedding of the desired step size... During training we... minimize a combined loss: (i) the standard score/flow matching loss... and (ii) a consistency loss...
- **Winner: B**. The DSD proposal with its consistency loss was deemed a more theoretically grounded and powerful approach for handling variable step sizes compared to SAFNet.

### Example 5: Stochastic Linear Bandits (ID: Oeb0I3JcVc)

- **Problem Statement:** Is it possible to derive instance-specific, data-driven regret guarantees for stochastic linear bandit algorithms... and to adjust these algorithms so that they preserve their empirical effectiveness while achieving the minimax optimal regret?
- **Agent A Response (`Mistral-24B` Self-Play):** We propose an adaptive, instance-specific regret-guaranteed stochastic linear bandit algorithm **ISRG-SLB** that dynamically adjusts its exploration-exploitation trade-off... ISRG-SLB employs a Bayesian posterior sampling mechanism...
- **Agent B Response (`Mistral-24B` → `GPT-OSS-120B`):** We introduce **Hardness-Adaptive Regret Guard (HARG)**, a meta-algorithm that runs a base linear bandit method... but continuously computes an instance-specific hardness score... When the score exceeds a threshold, HARG switches to a robust OFUL-style exploration policy...
- **Winner: C (Tie/Cannot Decide)**. The evaluator judged both proposals to be highly sophisticated and plausible meta-algorithms that addressed the core problem in different but equally valid ways, making a clear winner difficult to determine.

### Example 6: Root-Cause Analysis (ID: l11DZY5Nxu)

- **Problem Statement:** How can we reliably identify the true root-cause nodes of an anomaly in a complex system when anomalies are rare and lie outside the training distribution of a structural causal model, making counterfactual estimates unreliable?
- **Agent A Response (`Mistral-24B` → `GPT-OSS-120B`):** We introduce **Distribution-Aware Counterfactual Projection (DACP)**: (1) train a causal invertible generative model... on abundant normal operation data... (2) given an anomalous observation, compute its in-distribution projection... (3) for each candidate... perform an on-manifold do-intervention...
- **Agent B Response (`GPT-OSS-120B` → `Mistral-24B`):** We propose a novel approach called **Causal Influence Propagation (CIP)** that operates within the in-distribution regime by iteratively simulating the propagation of influences through the causal graph. Given an observed anomaly, CIP initializes a set of candidate root-cause nodes...
- **Winner: A**. The DACP proposal was judged to be a more rigorous and theoretically grounded approach by explicitly using a generative model to project anomalies back to the in-distribution manifold before intervention.

### Example 7: Promptable 3D Segmentation (ID: yXCTDhZDh6)

- **Problem Statement:** How can a unified, scalable foundation model be developed for promptable 3-D segmentation of point clouds that overcomes heterogeneous data formats and the scarcity of diverse, richly annotated 3-D mask data?

- **Agent A Response (`Mistral-24B` → `GPT-OSS-120B`):** We introduce **GeoPrompt-3D**, a unified sparse-voxel transformer that first tokenizes any point-cloud format into a geometry-aware token lattice... A multimodal prompt encoder maps textual descriptions, point clicks, or exemplar masks into a set of prompt tokens...

- **Agent B Response (Human Abstract):** The development of 2D foundation models for image segmentation has been significantly advanced by the Segment Anything Model (SAM). However, achieving similar success in 3D models remains a challenge... To this end, we propose a 3D promptable segmentation model Point-SAM... We then distill the rich knowledge from 2D SAM for Point-SAM training...

- **Winner: B (Human).** The human-proposed solution of distilling knowledge from the existing 2D SAM was judged to be a more pragmatic and powerful approach to overcoming data scarcity than the LLM's proposal of generating synthetic masks with a diffusion model.

# D EXTENDED QUALITATIVE ANALYSIS OF SOLUTION ARCHETYPES

To understand the conceptual landscape of the generated solutions, we performed a qualitative analysis by clustering the entire solution space. We grouped solutions into semantically coherent clusters via KMeans++ on their embeddings and assigned each cluster an interpretable label based on its most characteristic keywords.

This process reveals a rich taxonomy of 11 distinct solution archetypes, summarized in Table 8. The analysis demonstrates that the agents do not merely produce generic outputs but explore a diverse range of modern research paradigms. These archetypes span broad strategies like Reinforcement Learning (C0), foundational model architectures (C1), and highly specialized domains such as Molecular Graph Modeling (C2) and 3D Scene Representation (C8).

Table 8: Taxonomy of identified solution archetypes. Cohesion is the average intra-cluster cosine similarity.

| ID | Archetype Label | Top Keywords | Cohesion | Size |
|---|---|---|---|---|
| C0 | Reinforcement Learning | policy, learning, reward, agent, action | 0.48 | 1037 |
| C1 | Transformer Architectures | attention, model, token, rank, layer | 0.46 | 1106 |
| C2 | Molecular Graph Modeling | protein, graph, molecular, diffusion, equivariant | 0.48 | 398 |
| C3 | Neural/Latent Dynamics | neural, latent, time, network, dynamics | 0.41 | 899 |
| C4 | LLM Reasoning | model, llm, language, reasoning, human | 0.46 | 1647 |
| C5 | Adaptive Gradient Methods | gradient, adaptive, algorithm, convergence, learning | 0.38 | 936 |
| C6 | Multimodal Learning | visual, modal, language, transformer, video | 0.46 | 911 |
| C7 | Data-Centric AI | data, model, training, loss, class | 0.41 | 974 |
| C8 | 3D Scene Representation | 3d, scene, point, view, motion | 0.48 | 548 |
| C9 | Diffusion Models | diffusion, latent, image, model, diffusion model | 0.51 | 775 |
| C10 | Graph Neural Networks | graph, node, gnn, edge, graphs | 0.47 | 424 |

The conceptual cohesion of each archetype, measured by intra-cluster cosine similarity, offers further insight. For example, Diffusion Models (C9) form the most semantically coherent group (0.51 similarity), suggesting a well-defined and consistently applied solution pattern. In contrast, Adaptive Gradient Methods (C5) are the most diverse (0.38 similarity), indicating a broader set of varied approaches within that theme. This analysis confirms that the agents are capable of generating a wide spectrum of scientifically relevant and conceptually distinct solutions, providing a qualitative validation of their creative problem-solving abilities.

**Representative Solution Archetypes.** Below we provide a representative generated solution for each of the 11 identified clusters.

> **Cluster 0: Reinforcement Learning & Policy Optimization (N=1037)**
>
> **Keywords:** policy, learning, reward
> **Representative Solution:** We introduce **Meta-Regularized Adaptive Normalization (MRAN)**, a model-free actor-critic algorithm that augments a shared Transformer-based policy-value network with a lightweight context encoder.

**Cluster 1: Transformer Architecture & Attention (N=1106)**

**Keywords:** attention, model, token
**Representative Solution:** We introduce the **Meta-Optimized Sparse Mixture-of-Experts Transformer (MOS-MoE)**, in which a lightweight hypernetwork predicts, for each token and layer, a sparse subset (e.g., 2-4) of expert feed-forward networks.

**Cluster 2: Molecular & Protein Graph Learning (N=398)**

**Keywords:** protein, graph, molecular
**Representative Solution:** We propose a novel **Equivariant Variational Conformer Autoencoder (EVC-AE)** for structure-based drug design, which extends equivariant graph neural networks (EGNNs) with a variational autoencoder.

**Cluster 3: Neural ODEs & Continuous-Time Models (N=899)**

**Keywords:** neural, latent, time
**Representative Solution:** We propose a Temporal Neural Implicit ODE (TNI-ODE) architecture in which a deep network $f_\theta(x, t, \Delta t)$ parameterizes a continuous-time vector field that explicitly receives the elapsed time $\Delta t$ between observations.

**Cluster 4: Large Language Models & NLP (N=1647)**

**Keywords:** model, llm, language
**Representative Solution:** We introduce a hierarchical self-reward framework where a population of heterogeneous critic LLMs (parameter-perturbed copies) generate pairwise preference judgments for candidate completions, and a main LLM is fine-tuned on this preference data.

**Cluster 5: Optimization & Gradient Algorithms (N=936)**

**Keywords:** gradient, adaptive, algorithm
**Representative Solution:** We propose **Sparse Adaptive Sketch-Based Gradient Estimation (SASGE)**: at each online round the algorithm draws a set of $k = O(s \log d)$ *sparse* random perturbation vectors whose support follows a structured distribution.

**Cluster 6: Multimodal & Visual Language Models (N=911)**

**Keywords:** visual, modal, language
**Representative Solution:** We introduce a novel architecture called **Compositional Semantic Transformer (CoST)** that integrates a multi-modal encoder with a graph-based reasoning module. The CoST model encodes visual and linguistic inputs into a shared semantic space.

**Cluster 7: Adversarial Learning & Robustness (N=974)**

**Keywords:** data, model, training
**Representative Solution:** We propose a **Decoupled Adversarial Invariance Learning (DAIL)** framework that decouples the training of invariance and classification objectives using a dual-branch architecture with shared representations but separate adversarial heads.

**Cluster 8: 3D Vision & Scene Representation (N=548)**

**Keywords:** 3d, scene, point
**Representative Solution:** We introduce **Dynamic Implicit Neural Volume (DINV)**: given a few calibrated input images, a multi-scale patch transformer extracts per-patch embeddings and aggregates them via a cross-attention module to query a dynamic MLP-based radiance field.

**Cluster 9: Generative Diffusion Models (N=775)**

**Keywords:** diffusion, latent, image
**Representative Solution:** We propose a **Perceptual-Guided Diffusion (PGD) framework** that augments the standard denoising diffusion process with a trainable Perceptual Guidance Module (PGM). At each denoising step $t$, the cross-attention layers of the U-Net receive conditioning from the PGM.

---

**Cluster 10: Graph Neural Networks (GNNs) (N=424)**

**Keywords:** graph, node, gnn
**Representative Solution:** We introduce **Spectral-Diffusion Graph Transformers (SD-GT)**, which augment the standard self-attention with (i) a set of learnable wavelet-filtered Laplacian eigen-vectors that provide multi-scale spectral information and (ii) a graph diffusion process.

---

# E  AGENT AND CRITIC PROMPTS

This section contains the exact prompts used for the Generalizer agent, the Solver agent, and their corresponding internal/external critics.

---

**Prompt: Generalizer Agent ($\mathcal{G}$)**

**System Role:**

You are an AI researcher with 20 years of experience in the field. Your task is to read a research abstract and identify the core research question tackled by the paper. You must be extremely careful to:

- Preserve the fundamental scientific challenge
- Avoid hinting at specific solution methods
- Maintain precision and clarity

**User Prompt:**

**Original Research Abstract:**
{abstract}

**Your Task:**
Write the core research question that captures the core scientific problem described in the abstract.

**Requirements:**

- **Semantic Fidelity**: Preserve the fundamental scientific challenge exactly.
- **Information Preservation**: Retain all critical details, constraints, and insights.
- **Specificity**: Be precise and unambiguous.
- **Solution Blindness**: Do not hint at or describe the specific solution method.

**Output Format:**

- Provide your problem statement in 2-3 clear, concise sentences.
- Provide a justification for the identified research question.
- Enclose the problem statement inside `<problem_statement></problem_statement>` tags.
- Enclose your justification within `<justification></justification>` tags.

---

**Prompt: Generalizer Agent ($\mathcal{G}$)**

**System Role:**

You are an expert evaluator specializing in assessing the quality of problem statements from research abstracts. Your role is to critically evaluate whether a problem statement successfully captures the core idea of a research abstract across multiple dimensions.

**User Prompt:**

**Original Research Abstract:**
{original_abstract}

**Extracted Problem Statement to Evaluate:**
{problem}

**Evaluation Task:**
Assess the quality of the problem statement against the original abstract using the following dimensions:

- **Semantic Fidelity (1-10):** How well does the problem statement preserve the core scientific problem? (10 = identical challenge).
- **Information Loss (1-10):** Assess the severity of any missing critical details. (10 = critical info lost, 1 = no loss).
- **Ambiguity (1-10):** Rate the ambiguity of the problem. (10 = highly ambiguous, 1 = perfectly specific).
- **Solution Leakage (1-10):** Does it hint at the solution? (10 = explicitly describes solution, 1 = completely blind).

Finally, provide a final judgement on whether the problem statement preserves the core problem.

**Output Format:**
Provide your evaluation in a structured format with separate tags for each assessment (<semantic_fidelity_assessment>, etc.) and a <final_judgement> tag.

---

**Prompt: Solver Agent ($\mathcal{S}$)**

**System Role:**

You are an expert AI research scientist. Your task is to invent a plausible technical approach that could solve a given scientific problem in machine learning. You must be creative and innovative, proposing novel solutions.

**User Prompt:**

**Problem Statement:**
{problem}

**Your Task:**
Propose a specific and novel technical approach, mechanism, or architecture. Explain your proposed method in 3–5 sentences as if you're writing the core idea in a research paper.

**Requirements:**

- **Novelty & Creativity**: Propose a non-obvious, innovative solution.
- **Technical Feasibility**: Ensure your approach is logically sound and implementable.
- **Completeness**: Provide enough detail to understand the core methodology.

**Output Format:**

- Provide your proposed technical approach in 3-5 sentences.
- Provide a brief justification for the proposed solution.
- Enclose your solution inside <solution></solution> tags.
- Enclose your justification within <justification></justification> tags.

---

**Prompt: Solution Critic ( for Solver)**

**System Role:**

You are an expert evaluator specializing in assessing the quality of proposed solutions for complex research problems. Your role is to evaluate the proposed solution solely on how well it solves the problem. You must be thorough and objective in your assessment.

**User Prompt:**

**Problem Statement:**
{problem}

**Proposed Solution:**
{pred}

**Evaluation Task:**
Assess the quality of the proposed solution using the following dimensions:

- **Novelty & Creativity (1-10):** How novel is the approach? (10 = highly creative).
- **Technical Feasibility (1-10):** Is the solution technically sound? (10 = perfectly plausible).
- **Completeness & Detail (1-10):** How complete is the methodology? (10 = fully specified).

Finally, write your final judgement indicating whether the proposed solution ultimately solves the problem.

**Output Format:**
Provide your evaluation in a structured format with separate tags for each assessment (`<novelty_assessment>`, etc.) and a `<final_judgement>` tag.

---

