# OpenReview forum: "AInstein: Assessing the Feasibility of AI-Generated Approaches to Research Problems"
_ICLR.cc/2026/Conference — Submitted to ICLR 2026_

### Official Review · Reviewer_SCkB · 2025-10-28

**Soundness:** 1
**Presentation:** 2
**Contribution:** 2
**Rating:** 2
**Confidence:** 3

**Summary:**

This paper proposes AInstein, a framework for evaluating LLMs on the task of generating solutions to given research problems. AInstein focuses on evaluating LLMs in situations where they rely solely on their pretrained parametric knowledge, without using any external information. For evaluation, the authors create a benchmark consisting of problem–solution pairs extracted from the abstracts of ICLR 2025 papers. The evaluation is based on three criteria: success rate, rediscovery, and novelty, which are assessed using LLM-as-a-judge and cosine similarity of embeddings. The authors evaluate a solution-generation framework leveraging self-correction and claim that LLMs are capable of generating creative solutions rather than merely reproducing the human-generated solutions.

**Strengths:**

* This paper introduces a new benchmark for evaluating LLMs in generating solutions to given problem statements, which are extracted from ICLR 2025 papers. Although there are many studies on research idea generation, there is a lack of benchmarks that include reference solutions for given research questions. This dataset could serve as a useful benchmark for the research solution generation task.

**Weaknesses:**

* **The studied setting is not well-motivated.** This paper targets the setting of "using only their pretrained parametric knowledge," without relying on external aids. This is a reasonable baseline setting; however, it is not sufficiently justified to explore the setting without external information such as retrieval. The paper claims to "isolate scientific problem-solving ability," but the studied setting is largely affected by the amount of knowledge stored in the model parameters. If the goal is to isolate scientific problem-solving ability from other capabilities, it would be more reasonable to provide sufficient information necessary for solving the target problems.

* **Unreliable evaluation framework.** They employ cosine similarity between the problem and the generated solution to evaluate alignment. It is not sufficiently justified whether this simple metric effectively captures alignment quality. For example, demonstrating its correlation with human evaluation would strengthen the claim. Similarly, regarding LLM-as-a-judge, they state that "GPT-OSS-120B as an LLM-as-a-judge is known to correlate well with human evaluation," but they need to provide evidence of this specifically for the studied task.

---
Minor limitations

* **Findings are not surprising.** Although the settings are not exactly identical, many previous studies have shown that LLMs can generate creative and feasible research ideas [e.g., 1].

* **Novelty of the method.** The proposed framework is a straightforward application of self-correction to this task, which is also not novel.

* **Missing baselines.** This paper does not provide any baseline scores for comparison with LLM performance, such as human performance. As a result, from the provided numbers, it is difficult to understand how well LLMs perform on the proposed task.

[1] Si et al. (ICLR 2025). Can LLMs Generate Novel Research Ideas? A Large-Scale Human Study with 100+ NLP Researchers. https://openreview.net/forum?id=M23dTGWCZy

**Questions:**

I expect responses to the points I listed in the Weaknesses section.

---

> ### Author Response · Authors · 2025-11-27
>
> We thank the reviewer for their critical engagement with our work and for recognizing that our dataset is a useful benchmark due to the lack of available problem-solution pairs for research idea generation. We address the concerns regarding motivation, evaluation reliability, baselines, and novelty below.
>
> 1. **Isolating Parametric Knowledge**: Our central goal is to understand LLMs can genuinely come up with novel solutions on their own. Retrieval augmentation (RAG) fundamentally changes the problem by enabling the model to look up relevant documents, making the task "knowledge-augmented reasoning". Our framework, AINSTEIN, is specifically designed to isolate the ability to synthesize disparate observations into coherent theories using only internal knowledge, mirroring the human process of synthesizing internalized concepts. By removing external aids, we force the model to recombine its internal knowledge to re-derive or invent a method. This provides a clean measure of its capacity for conceptual synthesis, which is explicitly decoupled from data size or external resource access.
> 2. **Evaluation Framework Reliability**: Cosine similarity is used as a complementary quantitative metric to assess semantic coherence and textual alignment between problems and solutions. Our primary, high-fidelity metrics are the qualitative assessments provided by the LLM-as-a-Judge and the Human-Verified ELO Ranking. To validate the entire framework and address bias, we conducted a head-to-head tournament where human evaluators provided pairwise preferences between solutions, from which we computed an ELO rating for each agent configuration.
>    - The human-verified ELO ranking confirms the hierarchy produced by the LLM-as-a-Judge, where we validate a small fraction of LLM decisions to make sure that it is aligned.
>
>    - This ELO ranking serves as a robust, human-centric validation of the overall quality and ranking produced by our framework.
> 3. **Missing Baselines and Novelty**
>    - Novelty of Contribution: Our work introduces a new evaluation paradigm that isolates scientific problem-solving by decoupling problem extraction from solution generation. The key contribution is the systematic metric set - Success, Rediscovery, and Novelty - which allows for a finer-grained analysis of reasoning versus recall than previous studies. Previous works, such as the one cited, focus primarily on idea feasibility, whereas AINSTEIN measures divergence from established methods.
>    - Human Baseline Included: We included the Human Abstracts as the ultimate ground-truth baseline in the competitive evaluation. The ELO rating explicitly compares the generated LLM solutions against the human baseline. The Human Abstracts emerged as the clear top performer (ELO $1187$), confirming that the LLM task remains challenging.
>    - Methodological Novelty: While the use of self-correction loops is not new, our nested dual-critique framework (Internal $\mathcal{M}_{i}$ and External $\mathcal{M}_{e}$ critiques) is applied in a novel way to systematically emulate the scientific peer-review process for both problem abstraction and solution generation. The results explicitly show that the internal model's capability is paramount to success.

---

> > ### Comment · Reviewer_SCkB · 2025-11-27
> >
> > Thank you for your response. However, the authors’ reply does not directly address my concerns.
> >
> > * Isolating Parametric Knowledge
> >
> > The response does not address my concern that “the studied setting is largely affected by the amount of knowledge stored in the model parameters,” nor does it answer my suggestion that “If the goal is to isolate scientific problem-solving ability from other capabilities, it would be more reasonable to provide sufficient information necessary for solving the target problems.”
> >
> > * Evaluation Framework Reliability
> >
> > Your response does not directly answer my concerns about similarity-based and LLM-based evaluation.

---

> > > ### Author Response · Authors · 2025-11-28
> > >
> > > > the studied setting is largely affected by the amount of knowledge stored in the model parameters
> > >
> > > We agree with this statement. That is why we use highly-performant open models in our experiments that have been exposed to internet-scale of data with abundance of knowledge.
> > >
> > > > If the goal is to isolate scientific problem-solving ability from other capabilities, it would be more reasonable to provide sufficient information necessary for solving the target problems.
> > >
> > > The underlying assumption of our study is that there is enough knowledge encoded in a decent LLM's parameters to answer most scientific queries. LLMs are not trained to establish connections between different concepts in its knowledge space for a given task by default, but they are good instruction followers, and prior works show that if we present the solution recipe to LLMs describing what concepts to connect and how to use it (aka instructions), they are capable of doing it. In this work, we explore if LLMs can be used in a similar fashion to solve research problems except we want the LLM to discover those recipes by itself. This is the fundamental research setting we are operating in. Connecting AInstein to a RAG system can help, but our current focus is to study how far we can go with just the model's internal knowledge. So we are unsure if this is necessary to explore the RAG setting, as it is also challenging to benchmark AInstein with RAG while ensuring that the model doesn't cheat to propose the answer (as our research questions are constructed from paper that are indexed by all major academic databases). Additionally, using a RAG system also comes with extra costs, constraints, and bottlenecks, so our research has positive impacts there too. For instance, instead of invoking a RAG call for each query, we only use it when really necessary. Other examples of more common task with a similar flavor are that of auto-routing input queries to an LLM, where the router decides whether to use the most expensive variant of the model for the query or a cheaper, faster variant to answer a query. Another similar example is that of chatGPT, where we use an intelligent mechanism to reduce reliance on external web search -- for topics like current affairs, we cannot avoid it, but for conceptual queries in science, it can definitely be reduced.
> > >
> > > > Evaluation Framework Reliability. Your response does not directly answer my concerns about similarity-based and LLM-based evaluation.
> > >
> > > We acknowledge that there limitations to similarity-based and LLM-based evaluations, but our current approach of validating a small fraction of LLM judgements to ensure its alignment is a valid technique used in prior work (Liang et al., 2024). That being said, we are preparing a larger-scale human evaluation to better establish the soundness of our methodology.
> > >
> > > We are sincerely grateful for your interest and feedback for our work, and are happy to address any other queries you might have.
> > >
> > > Thank you for continued engagement.
> > >
> > > Liang et al., (2024). Can Large Language Models Provide Useful Feedback on Research Papers? A Large-Scale Empirical Analysis. NEJM. https://ai.nejm.org/doi/full/10.1056/AIoa2400196

---

### Official Review · Reviewer_nyGo · 2025-10-30

**Soundness:** 3
**Presentation:** 3
**Contribution:** 3
**Rating:** 6
**Confidence:** 4

**Summary:**

This paper investigates whether large language models can serve as autonomous scientific problem-solvers. It introduces an iterative proposal–critique–revision framework that includes both internal and external critique loops to extract research problems from existing ICLR 2025 papers while minimizing information leakage and solution bias. To evaluate solution quality, the authors propose three complementary metrics: Success Rate, Rediscovery, and Novelty. They combine an LLM-as-a-judge rubric with selective human verification through a head-to-head tournament. The results show that solution quality is closely tied to the ability of the reasoning model, and that current LLMs are more adept at generating novel yet scientifically valid alternatives than at reproducing human-designed methods.

**Strengths:**

This paper explores a highly interesting and valuable question about whether large language models can serve as reliable autonomous problem solvers for research problems. It examines the originality and divergence of LLM-generated solutions compared with human-designed ones.

The paper presents an iterative proposal, critique, and revision framework that enables accurate problem extraction without solution leakage while improving the quality of generated solutions.

It also introduces systematic evaluation metrics, including success, rediscovery, and novelty, which allow for a fine-grained analysis to provide clear insights into the capabilities and limitations of current LLMs in scientific problem solving.

**Weaknesses:**

**Overclaim about question:** Although this paper claims to explore whether LLMs can solve AI research problems, the actual experiments only investigate the generation of solution proposals, rather than their detailed implementation. As we know, even if two solution descriptions appear similar, the specific implementation details can lead to drastically different outcomes. Therefore, the act of “solving a problem” should not be separated from its implementation — they should be evaluated as an integrated whole. From this perspective, the paper does not convincingly answer its central question.

**Potential Data Leakage:** While the paper states that “to prevent data leakage and ensure our evaluation tests reasoning rather than retrieval, all models used in our experiments have knowledge cutoffs that predate the ICLR 2025 submission deadline”, I believe data leakage may still exist. Many ICLR papers are resubmissions of earlier works, meaning these papers are probably already available on the internet before the ICLR 2025.  For instance, although GPT-OSS-120B has a knowledge cutoff in June 2024, many papers were already online several months earlier, especially resubmissions, making it possible that some of this information was included in the model’s training data.

**Questions:**

1: When computing the Elo scores based on human evaluations, were the comparisons conducted across different human evaluators? Human bias (e.g., differences in background knowledge or familiarity with specific research areas) could strongly affect the preference judgments between different solutions.

2: How does the number of iterative refinement cycles in the critique loops influence the final results, such as the success rate? Have you explored how performance varies with different numbers of refinement iterations?

3: Have you tried retrieval-augmented approaches, where the model retrieves related papers or prior works to inspire more accurate or innovative solutions? I believe incorporating retrieval could potentially improve both the quality and originality of the generated research proposals.

---

> ### Author Response · Authors · 2025-11-27
>
> We thank the reviewer for their positive assessment, recognizing the highly interesting question and the utility of our iterative proposal, critique, and revision framework with its systematic metrics. We address the weaknesses and questions raised below.
>
> 1. **"Solving" vs. "Generating Proposals"**: We agree that a complete solution requires implementation. Our goal, however, is to isolate and measure the latent reasoning ability, the capacity to propose a technically plausible approach that, in principle, could be implemented. This mirrors the crucial idea generation stage of the scientific process. We acknowledge that implementation is the natural next step, and we defer a downstream implementation benchmark (e.g., code synthesis and execution) to future work.
> 2. **Potential Data Leakage**: All models used have a knowledge cutoff that predates the ICLR 2025 submission deadline. Even if slight leakage occurred, the Problem Extraction Phase is designed to remove explicit solution cues and artifacts. The success of the Solver in generating a Novel & Valid approach (i.e., one that doesn't match the original) confirms the model is engaging in reasoning beyond simple recall.
> 3. **ELO Scores and Human Bias (Question 1)**
> The ELO rating was computed from pairwise preferences collected within our research team , where solutions were presented in a randomized and anonymized fashion. This process served as a high-fidelity internal validation to confirm the findings of the automated judges. To address the need for external, diverse expertise, we are planning a larger-scale human study involving external domain experts to validate the framework's generalizability and reliability.
> 4. **Influence of Critique Cycles (Question 2)**: We did not see any major trends in performance v/s number of refinement cycles (at least for n values less than 20). Instead we find that a critical factor for success is the capability of the internal model ($\mathcal{M}_{i}$). We select N=20 maximum attempts to ensure convergence across all model configurations.
> 5. **Retrieval-Augmented Approaches (Question 3)**: Our core aim is to test reasoning using parametric knowledge only. Retrieval augmentation changes the question to "knowledge-augmented reasoning," which is an orthogonal line of inquiry.

---

### Official Review · Reviewer_SRkR · 2025-10-31

**Soundness:** 3
**Presentation:** 3
**Contribution:** 3
**Rating:** 6
**Confidence:** 2

**Summary:**

This paper introduces a new framework to test whether large language models (LLMs) can act as independent scientific problem-solvers using only their internal knowledge without fine-tuning, retrieval, or external data. The proposed AInstein framework mimics the real research cycle through two stages: Problem Extraction and Solution Generation. Using 1,214 ICLR papers across Oral, Spotlight, and Poster tiers, the study measures Success Rate, Rediscovery, and Novelty with both LLM and human evaluation. Results show that powerful models (e.g., GPT-OSS-120B) can rediscover feasible ideas and even generate new, valid research directions, though their reasoning remains fragile and sensitive to prompt framing.

**Strengths:**

- The paper raises a novel and important research question: whether LLMs can independently solve research problems without any external knowledge.
- The two-phase setup (Problem Extraction + Solution Generation) with internal/external review loops nicely reflects how real research and peer review work.
- The experimental setup is solid. The dataset includes 1,214 ICLR 2025 papers across multiple acceptance tiers (Oral, Spotlight, Poster), and multiple model families are tested.

**Weaknesses:**

- The study focuses only on AI/ML papers, so it’s uncertain whether the findings generalize to other fields such as biology or physics.
- Even though the models were trained before ICLR 2025, they might still have internalized common research patterns, making it difficult to fully separate reasoning from memorization.
- The evaluation remains at the text level. There’s no actual implementation or experimental validation to confirm whether the generated solutions would work in practice.

**Questions:**

- Can the findings generalize beyond the AI/ML domain to other scientific domains?
- How can we truly determine whether the model is memorizing familiar research patterns rather than demonstrating genuine reasoning?
- How can we be sure that the generated solutions would actually work in practice?

---

> ### Author Response · Authors · 2025-11-27
>
> We thank the reviewer for their positive assessment, recognizing the novelty of our research question , the solid experimental setup, and the appropriate mirroring of the research workflow through the critique loops. We address the weaknesses regarding generalization, memorization, and implementation below.
> 1. **Generalization**:
>       - Framework Applicability: We agree that generalization is crucial. The AINSTEIN pipeline is fundamentally domain-agnostic; it requires only a scientific abstract (source $\mathcal{A}$) to produce a problem statement ($\mathcal{P}$) and a solution ($\mathcal{Z}$).
>       - Future Work & Plan: We explicitly state that future work should extend this framework to other scientific fields like biology and physics. The same methodological principles apply across scientific fields, and the only required adaptation is a refined prompt for the Generalizer agent to handle domain-specific terminology.
> 2. **Disentangling Memorization vs. Reasoning**
>      - Our study is designed to isolate the ability to generate solutions using only parametric knowledge without retrieval augmentation or fine-tuning. This controlled design is precisely intended to probe genuine reasoning, as opposed to sophisticated pattern matching or recall.
>
>       - Novelty Metric Quantifies Reasoning: We directly quantify non-memorized reasoning through the Novelty metric: solutions that are valid but different from the original human paper. The model's consistently high score on the Novel & Valid metric (e.g., $57-65\%$ under strict $\tau=5$ evaluation, Table 3) provides large-scale evidence that they generate original, sound alternatives, going beyond simply regurgitating familiar patterns.
> 3. **Lack of Implementation / Empirical Validation**: Could you please further clarify on this point? We have conducted extensive empirical evaluations in our work, so it would be helpful if you could let us know in more detail what you felt lacking in implementation.

---

### Official Review · Reviewer_TMVU · 2025-10-31

**Soundness:** 1
**Presentation:** 3
**Contribution:** 2
**Rating:** 2
**Confidence:** 5

**Summary:**

This paper introduces AInstein, a framework for evaluating whether LLMs can autonomously generate viable research solutions without external knowledge. The system distills problem statements from recent ICLR papers and uses nested critique loops to produce and iteratively refine methods, which are then assessed through automated scoring and human comparison.

**Strengths:**

1, The problem studied int this work  is timely and ambitious;

2, The proposed problem-to-solution and dual-critique framework is well designed and mimics real scientific workflows.

3, The experiments cover a broad range of topics and models, and reported multi-dimensional analyses (e.g., rediscovery, novelty, human comparison).

**Weaknesses:**

1, "Feasibility" may need to consider compute constraints、data acquisition practicality.
2,  A big concern when comparing with human-written methods: LLMs or human annotators may identify human-authored solutions from academic phrasing and reward them unfairly. Incorporating style obfuscation or content-only evaluation would strengthen the fairness of human-vs-model comparisons.
3, The paper's human verification is limited to the authors evaluating their own system's outputs. This is a major methodology flaw: the risk of self-evaluation bias is high, and given the breadth of ICLR topics, the authors are unlikely to be qualified to judge novelty and technical correctness across all problem domains. Without external and diverse experts, the setup here is not acceptable.

**Questions:**

N/A

---

> ### Author Response · Authors · 2025-11-27
>
> We thank the reviewer for their careful reading and for recognizing the ambition and design quality of the AINSTEIN framework and the dual-critique mechanism. We address the points raised below.
> 1. **Feasibility**: The framework's goal is to isolate the innate reasoning ability in LLMs. We evaluate Technical Feasibility (is the solution logically sound and implementable in principle) and not resource management. Moreover, The full pipeline cost is low as we utilize open source LLMs, and thus, total costs for the 1,214 papers are within typical research budgets.
> 2. **Fairness and Academic Phrasing Leakage**:
>       - Mitigation through Problem Extraction: Our two-phase design explicitly addresses this. The     Generalizer agent is strictly instructed to maximize abstraction and minimize Solution Leakage , ensuring the Solver sees only the distilled problem P, a few examples are also showcased in the appendix for reference.
>       - Focus on Novelty Confirms Content: Our results already show that LLMs excel at generating Novel & Valid solutions, confirming they are inventing sound alternatives, not recalling style. We observe a stark drop in Rediscovery rates from the lenient ($\tau=4$) to strict ($\tau=5$) evaluation (Table 3), which reveals that perfect retrieval/pattern matching is exceptionally rare.
> 3. We conducted robust internal validation using the Human-Verified Competitive Ranking (ELO, Section 5). This ranking was computed from pairwise preferences collected within our research team , where solutions were presented in a randomized and anonymized fashion. This process was intended as a high-fidelity internal validation to confirm the findings of the automated judges. To address the need for external, diverse expertise, we are planning a larger-scale human study involving external domain experts to validate the framework's generalizability and reliability.

---

### Official Review · Reviewer_4xgZ · 2025-11-01

**Soundness:** 2
**Presentation:** 3
**Contribution:** 2
**Rating:** 4
**Confidence:** 4

**Summary:**

In this paper, the authors propose AInstein, a framework for evaluating whether LLMs can generate valid solutions to AI research problems using only their pretrained parametric knowledge. They first collect the accepted papers from ICLR 2025 and rewrite the abstract to become a question. Then, an iterative question-solving method is used to generate an answer. Finally, these results are compared with human solutions with LLM-as-a-Judge. Results show that the problem-solving ability of LLM remains fragile and highly sensitive to framing.

**Strengths:**

1. AI4Research is a very important topic, and it might be shaping the next generation of AI.
2. The proposed method and datasets are intuitive and easy to use.
3. The paper is relatively easy to read, and Figure 1 is helpful in understanding the logic.

**Weaknesses:**

1. I am trying to convince myself about the motivation. Why do we need to isolate LLM from external knowledge searching? In my opinion, this is a normal step for human researchers. The authors mention: "(a) synthesize disparate observations into coherent theories and (b) traverse the conceptual path from problem to principle to solution". However, these things can be tested with resource searching as well. Besides, LLMs are trained on most of the data provided on the Internet; there is no difference between using the external knowledge or not if the knowledge cutoff is the same.

2. Lacks human evaluation. First, all experiments are evaluated by LLM-as-a-Judge, we do not know the accuracy of LLM judge, and research points out that the LLM judge is biased towards its own kind. Second, the metrics themselve are newly proposed, and we do not know the accuracy of the metrics. Overall, at least some human annotation is needed to compute the similarity between human and LLM annotations to make the annotation useful.

3. Knowledge cutoff is unsure. It is not known whether the model knows the provided ICLR 2025 paper and generates the answer from memory.

**Questions:**

See above.

---

> ### Author Response · Authors · 2025-11-27
>
> We thank the reviewer for their thoughtful comments and for recognizing the importance of AI4Research and the intuition behind our framework, AINSTEIN. We address the questions regarding motivation, evaluation, and knowledge cutoff below.
> 1. **Motivation: Isolating Scientific Reasoning vs. Information Retrieval** - The decision to isolate the LLM from external knowledge is central to our scientific question : Can LLMs solve AI research problems using only their parametric knowledge? We aim to measure the ability to synthesize concepts (reasoning), not just recall. Allowing retrieval would conflate high scores with memorization, which doesn't answer our question about autonomous problem-solving.  The Problem Extraction Phase deliberately removes solution cues, ensuring success in the Solution Phase must arise from re-derivation or invention. When a researcher solves a new problem, they typically do not look for a ready‑made solution; instead they synthesize concepts that they have internalised over years of study.  We emulate this by removing any retrieval or fine‑tuning that could serve as a shortcut to the original paper.
> 2. **Human Evaluation and Robustness Checks** -
> We recognize the concerns about using LLM-as-a-Judge, which is why our methodology includes several human-verified steps.
>      - Scalability: We evaluated 1,214 problem-solution pairs, making exhaustive human annotation challenging.
>      - Targeted Verification: Our evaluation includes an LLM-as-a-judge paradigm guided by a structured rubric, complemented by targeted manual checks as shared in the appendix with concrete examples.
>       - Robustness Checks: We confirmed our qualitative conclusions by using two independent LLM judges (GPT-OSS-120B and Qwen-235B). Furthermore, we conducted a human-verified head-to-head tournament on a representative subset to establish a definitive quality hierarchy. The GPT-OSS-120B self-play configuration was the clear top performer among LLMs (ELO = 1119).
>
> 3. **Knowledge Cutoff** -
> All models used (GPT-OSS-120B, Qwen-235B, Mistral-24B) have knowledge cutoffs that predate the ICLR 2025 submission deadline. This ensures we test reasoning, not retrieval of the original paper's solution to the best of our abilities.

---

### Meta-Review · Area_Chair_Xxu2 · 2026-01-05

**Summary:**

The submission "AInstein: Assessing the Feasibility of AI-Generated Approaches to Research Problems" evaluates LLMs (and systems built with them using internal critique) on the task of generating a plausible-sounding abstract given short problem statements sourced from ICLR 2025 accepted papers. The authors argue that this approach evaluates the system's capability of acting as autonomous scientific problem-solvers.

**Reviewer Concerns:**

Reviewers are concerned with several problems:
1. Motivation: It is unclear to several reviewers whether the capability that is tested here is related to scientific problem-solving, and more concretely to the 'feasibility' of AI-generated approaches. The model is trained only to 'generate proposal' and without external information, for example provided through search or offline retrieval, the task seems to underspecified and the motivation unclear.

    For example, given the following (unabriged) problem statement from the paper:

    > How can a unified, scalable foundation model be developed for promptable 3-D
    segmentation of point clouds that overcomes heterogeneous data formats and the scarcity of diverse,
    richly annotated 3-D mask data?

    The evaluated models provide only a short proposal, which is in turn only evaluated by another (relatively small) model. At no point is the proposal evaluated with sufficient precision to verify whether it is a feasible proposal toward a solution to a given problem (and there is not enough info in the problem statement to fully specify the problem), or a collection of words that 'mimicking' such a proposal.


2. Data Leakage. The authors use papers from ICLR 2025 without filtering. This introduces the potential for data leakage in two ways. First, not all submission to ICLR 2025 are novel, and have been available online for unspecified amounts of time as preprints. Second, be cause problem statements are short and relatively generic, the model may recall 'common research patterns' for these problems that are strung together from earlier works on the problem. The reviewers raised this issue during discussion, but the authors have chosen not to filter for either concern.

3. No impact from agent setup. The submission argues that the tested iterative setup is related to human review cycles. However, the authors also point out that performance of this iterative system is not actually related to its iterations "We did not see any major trends in performance v/s number of refinement cycles (at least for n values less than 20)."

4. Grounding. There is very little grounding for the results of the paper. The merit of the generated proposals is only evaluated by the same class of LLM with similar instructions, as reviewer SCkB notes 'The paper's human verification is limited to the authors evaluating their own system's outputs'. The submission claims that GPT-OSS-120b is a strong judge model, but only cites papers that evaluate older GPT-4 models and does not validate this claim in-domain. The submission includes a small human study, but it is unclear whether human evaluators had sufficient time and expertise to judge proposals. The submission further measures embedding similarity between generated and original proposal, but it seems unclear how this is related to the scientific quality or novelty of the generated proposal.

5. Novelty. The related work section relates to a number of generic topics that are not strongly related to the problem investigated in this work. It would be better to discuss related work such as
    * Gupta&Pruthi "All That Glitters is Not Novel: Plagiarism in AI Generated Research",
    * Gottweis, "Towards an AI co-scientist"
    * Si et al. "Can LLMs Generate Novel Research Ideas? A Large-Scale Human Study with 100+ NLP Researchers"

**Reviewer Scores:**

4xgZ remain at 4;
TMVU likely not to respond, remain at 2;
SRkR remain at 6;
nyGo remain at 6, data leakage issue not addressed;
SCkB remain at 2, unconvinced after author clarification

---

### Decision · Program_Chairs · 2026-01-26

Reject